# The Ratio of Baseline Ventricle Volume to Total Brain Volume Predicts Postoperative Ventriculo-Peritoneal Shunt Dependency after Sporadic Vestibular Schwannoma Surgery

**DOI:** 10.3390/jcm13195789

**Published:** 2024-09-28

**Authors:** Lisa Haddad, Franziska Glieme, Martin Vychopen, Felix Arlt, Alim Emre Basaran, Erdem Güresir, Johannes Wach

**Affiliations:** 1Department of Neurosurgery, University Hospital Leipzig, 04275 Leipzig, Germany; 2Comprehensive Cancer Center Central Germany, Partner Site Leipzig, 04103 Leipzig, Germany

**Keywords:** vestibular schwannoma, brain volumetrics, postoperative hydrocephalus, ventriculo-peritoneal shunt dependency

## Abstract

**Background/Objectives:** Obstructive hydrocephalus associated with vestibular schwannoma (VS) is the most common in giant VS. Despite tumor removal, some patients may require ongoing ventriculo-peritoneal (VP) surgery. This investigation explores the factors contributing to the requirement for VP surgery following VS surgery in instances of persistent hydrocephalus (HCP). **Methods:** Volumetric MRI analyses of pre- and postoperative tumor volumes, cerebellum, cerebrum, ventricle system, fourth ventricle, brainstem, and peritumoral edema were conducted using Brainlab Smartbrush and 3D Slicer. The total brain volume was defined as the sum of the cerebrum, cerebellum, and brainstem. ROC analyses were performed to identify the optimum cut-off values of the volumetric data. **Results:** Permanent cerebrospinal fluid (CSF) diversion after surgery was indicated in 12 patients (12/71; 16.9%). The ratio of baseline volume fraction of brain ventricles to total brain ventricle volume (VTB ratio) was found to predict postoperative VP shunt dependency. The AUC was 0.71 (95% CI: 0.51–0.91), and the optimum threshold value (</≥0.449) yielded a sensitivity and specificity of 67% and 81%, respectively. Multivariable logistic regression analyses of imaging data (pre- and postoperative VS volume, VTB ratio, and extent of resection (%) (EoR)) and patient-specific factors revealed that an increased VTB ratio (≥0.049, OR: 6.2, 95% CI: 1.0–38.0, *p* = 0.047) and an EoR < 96.4% (OR: 9.1, 95% CI: 1.2–69.3, *p* = 0.032) were independently associated with postoperative VP shunt dependency. **Conclusions:** Primary tumor removal remains the best treatment to reduce the risk of postoperative persistent hydrocephalus. However, patients with an increased preoperative VTB ratio are prone to needing postoperative VP shunt surgery and may benefit from perioperative EVD placement.

## 1. Introduction

Vestibular schwannoma (VS) is a benign tumor of the vestibular branch of cranial nerve VIII with an incidence of approximately 1–2/100.000 per year, accounting for 6–7% of all intracranial tumors [1].

Diagnosis is made through a magnetic resonance imaging (MRI) scan. This typically reveals a mostly homogeneous intrameatal tumor with contrast enhancement, which may extend into the cerebellopontine angle.

In addition to typical clinical signs such as unilateral hearing loss and tinnitus, large VS can compress the fourth ventricle, brainstem, and cerebellum [2,3]. Consequently, obstructive hydrocephalus is more commonly associated with large VS [4].

Symptomatic hydrocephalus occurs in 3.7–42% of these cases [5,6,7,8,9,10,11]. Previous studies have already investigated the potential factors predicting the persistence of hydrocephalus after VS surgery. Suggested risk factors include age [8], tumor size [11,12], a long course of disease [13], irregular tumor surface [4], the presence of severe hydrocephalus (HCP) before surgery [10], the presence of a postoperative hematoma volume > 10 cc [14], and a higher protein concentration in CSF [11].

Despite tumor removal being the optimal therapy for preventing hydrocephalus [10,15,16], a proportion of patients still develop hydrocephalus postoperatively, even after near or gross total resection.

The volume of the ventricles, VS, and brain structures being compressed by VS have not been considered as possible predictors for the occurrence of postoperative hydrocephalus so far. Nevertheless, volumetric measurements have previously been used to predict regression in patients with residual tumor [17] or to predict hearing preservation after surgery [18].

This study examines the potential of volumetric imaging parameters along with patient-, disease-, and treatment-specific factors as predictors of postoperative hydrocephalus following VS surgery in a semi-sitting position via the retrosigmoid approach. The aim is to identify those patients who might benefit from temporary perioperative external ventricular drain (EVD) placement in order to prevent persistent hydrocephalus.

## 2. Materials and Methods

### 2.1. Study Design and Patient Characteristics

One hundred and twenty patients with sporadic VS who underwent consecutive surgery between 2012 and 2022 were screened for inclusion in this retrospective study. Inclusion criteria were defined as follows: (1) histopathologically confirmed VS, (2) patients older than 18 years, and (3) Koos grades 1–4.

Patients with missing pre- or postoperative gadolinium-enhanced T1-weighted MRI data (*n* = 36), hemorrhage (*n* = 7), or meningitis (*n* = 6) were excluded. Patients with known neurofibromatosis type II were not included due to their different neuropathology [19].

Further analyses were performed on 71 patients, including patient data, pre- and postoperative volumetric MRI analysis, and clinical and histopathological data.

### 2.2. Clinical Data Recording

Furthermore, data were collected regarding the occurrence of hydrocephalus with VP shunt placement in cases without postoperative hematoma or meningitis, Koos grade, and Molecular Immunology Borstel I (MIB-I) Index (%) [20].

Clinical data of the patients were also obtained: age, sex, comorbidities, body mass index (BMI), surgical position, and pre- and postoperative cranial nerve function of nerves V, VII, VIII, and IX. For the facial nerve, data were gathered during the clinical stay and after 3 months, 12 months, and 24 months (using the House–Brackmann scale). Only patients who underwent VP-Shunt surgery within 30 days after VS surgery were included in this study.

### 2.3. Image Analyses

The segmentation to assess volumetric parameters was performed using Brainlab Smartbrush (BrainLAB AG, Feldkirchen, Germany), and surface area was assessed using 3D Slicer (Version 5.2.1, Surgical Planning Laboratory, Harvard University, Cambridge, MA, USA). The volumetric data of the cerebrum, cerebellum, ventricle system, and brainstem were created with the help of preset selection and were controlled manually for wrong identification (*n* = 0). The cerebrum was defined as the cerebral cortex and the subcortical structures including the hippocampus, basal ganglia, and amygdala. The cerebellum comprised the cerebellar hemispheres and vermis.

The ventricle system encompassed the inner cerebrospinal fluid space with the lateral ventricles, third ventricle, and fourth ventricle, but not the outer cerebrospinal fluid space.

Total brain volume was defined as the sum of the cerebrum, cerebellum, and brainstem. Tumor volume was not included in the measurement of total brain volume. The fourth ventricle, tumor, and edema were selected and measured manually for each patient. The fourth ventricle included the space between the pons and medulla oblongata, cerebellar peduncles, and inferior and superior medullary velum, but not the cerebral aqueduct. The extent of resection in % (EoR) was volumetrically determined by segmentation of the tumor volumes in pre- and postoperative 1.5 or 3.0 Tesla Gd-enhanced T1-weighted MR images. For exemplary data of the segmentation, see Figure 1. The Evans index was measured manually to assess potential differences between shunt-requiring and non-shunt-requiring patients. A ratio of baseline volume fraction of the brain ventricles to total brain volume (VTB ratio) was calculated [21].
$$\mathrm{VTB}\;\mathrm{ratio}\;=\frac{ventricle\;system\;volume\;(cm^{3\;})}{total\;brain\;volume\;(cm^{3})}$$

### 2.4. Surgical Procedure and Follow-Up Regime

Neuronavigation-guided resection (Brainlab Curve, BrainLAB AG, Feldkirchen, Germany) was performed under general anesthesia via the retrosigmoid approach in the semi-sitting position. Microscopic resection was performed using an operating microscope (Pentero, Carl Zeiss, Jena, Germany).

Clinical and imaging follow-up appointments included an MRI taken at 3 months after VS surgery. Earlier examinations were scheduled in cases of new or worsening neurological deficits or neuroradiological signs of VS progression. Follow-up MR images were interpreted by independent radiologists. Treatment decisions were made in an interdisciplinary neuro-oncological board.

### 2.5. Immunohistochemistry

Hematoxylin and eosin coloring was performed on the tissue samples. Then, immunohistochemical analysis was followed using antibodies like MIB-I (Dako, Glastrop, Denmark) for the detection of the Ki67 antigen and antibodies to S-100 (Dako, Glastrop, Denmark). The MIB-I Index offers the possibility of making a prediction about the proliferation of neoplastic cells by estimating the percentage of colored nuclei among all visible nuclei per high-energy field [20] and was also evaluated.

### 2.6. Statistics of Institutional Data

Data were dichotomized for the volume of VS, ventricle system, and cerebellum by using receiver operating characteristic (ROC) curve analysis. Fisher’s exact tests (two-sided) were utilized to compare categorical data, while Student’s *t*-test was employed for metric variables across the investigated patient groups (postoperative VP shunt dependency or not). Optimal cut-off values for metric variables in the prediction of postoperative VP shunt dependency were analyzed using ROC curve analysis. Multivariable binary logistic regression analysis of predictors for VP shunt surgery was performed. Univariable and multivariable tests were performed in SPSS (version 29 for Windows, IBM Corp., Armonk, New York, NY, USA). Graphs were created using Python 3.0. in the Jupyter Notebook application.

## 3. Results

### 3.1. Patient Characteristics

The data from 71 patients who suited the inclusion requirements were examined. The patient population had a predominance of women (56.3% women vs. 43.7% men), with a median age (+/−SD) of 56.9 years (+/−15.5 years).

The VS had a mean volume of 3.54 +/− 12.39 cm^3^ (+/−SD). Four patients (5.6%) suffered preoperatively from hydrocephalus; none of them underwent preoperative VP shunting. Thirty-four cases (47.9%) had peritumoral brain edema, with a mean (+/−SD) edema volume of 0.25 cm^3^. One patient (1.4%) had Koos grade 1; thirteen patients (18.3%) had Koos grade 2; seventeen patients (23.9%) had Koos grade 3; and forty patients (56.3%) had Koos grade 4. The mean (+/−SD) MIB-I Index (%) was 3.0 +/− 1.24. Preoperative dysfunctions of the cranial nerves were noted; 14 patients (19.7%) had preoperative dysfunction of the V cranial nerve and 14 patients (19.7%) had preoperative dysfunction of the VII cranial nerve. Preoperative dysfunction of cranial nerve VIII was observed in 71 cases (100%), hypacusis in 59 cases (83.1%), and anacusis in 12 cases (16.9%). In four patients (5.6%), dysphagia (IX cranial nerve) was found. Further details are provided in Table 1.

### 3.2. Patient Characteristics in VP-Shunt-Dependent and Non-Dependent Patients

A total of 12 patients (16.9%), of which 25% were men, required postoperative VP-shunt insertion. Patients with shunt dependency had an EoR of 81.49% +/− 28.344 (*p* = 0.146). Patients with larger ventricle systems (>52.15 cm^3^) accounted for 66.7% (*p* < 0.05) of those requiring shunting. No significant differences in the Evans index were observed between shunt-requiring and non-shunt-requiring patients (*p* > 0.05).

The descriptive statistics for the volumetric data of the measured structures are listed in Table 2.

The VTB ratio showed significant correlations with the shunt dependency (*p* < 0.001); the results are visualized for the two groups of shunt-dependent and non-dependent patients (see Figure 2 for boxplots).

### 3.3. Volumetrics and Association with Ventriculo-Peritoneal (VP) Shunt Dependency

ROC curve analyses were used to identify the optimum cut-off values for the volumes of the cerebrum, cerebellum, brain stem, ventricle system, fourth ventricle, edema, tumor, and tumor residue in mm (axial), as well as the VTB ratio, in predicting VP-shunt-dependent postoperative hydrocephalus.

The area under the curve (AUC) for the volume of the cerebrum in predicting VP shunt placement was 0.57 (95% CI: 0.42–0.72, *p* = 0.44), and, for the volume of the brainstem, it was 0.60 (95% CI: 0.44–0.77, *p* = 0.26).

The AUC for the volume of the ventricle system and for the fourth ventricle in predicting VP shunt placement were 0.68 (95% CI: 0.486–0.892, *p =* 0.068) and 0.53 (95% CI: 0.30–0.76, *p* = 0.75). The optimum cut-off value of ventricle volume was ≥52.15 cm^3^, with a sensitivity and specificity of 66.7% and 81.4%, respectively. For the cerebellum, the optimum cut-off value was at ≤110.5 with a sensitivity of 66.7% and a specificity of 81.4%. The AUC for the volume of the peritumoral edema in predicting VP shunt placement was 0.54 (95% CI: 0.30–0.77, *p* = 0.77). For the tumor volume, the AUC was 0.572 (95% CI: 0.385–0.760, *p* = 0.44); for the residual tumor volume, it was 0.518 (95% CI: 0.305–0.731, *p* = 0.87); and for the EoR (%), it was 0.614 (95% CI: 0.40–0.83, *p* = 0.30).

The area under the curve (AUC) for the VTB ratio in predicting VP shunt placement was 0.712 (95% CI: 0.51–0.91, *p* = 0.036). The optimal cut-off was ≥0.449, with a sensitivity of 67% and a specificity of 81%. For the ROC analyses of the VTB ratio and the occurrence of hydrocephalus, see Figure 3.

### 3.4. Multivariable Analysis of Risk Factors for Persistent Hydrocephalus

A multivariable analysis was conducted for the variables VTB ratio, EoR (%), preoperative tumor volume, residual tumor volume, and the occurrence of preoperative hydrocephalus to identify the risk factors for persistent hydrocephalus. Figure 4 shows forest plots summarizing the results of the multivariate binary logistic regression analysis.

An increased VTB ratio (≥0.049, OR: 6.2, 95% CI: 1.0–38.0, *p* = 0.047) and an EoR < 96.4% (OR: 9.1, 95% CI: 1.2–69.3, *p* = 0.032) were identified as independent predictors of postoperative VP shunt dependency.

## 4. Discussion

This study examined the preoperative factors potentially predicting the need for VP surgery after VS surgery in the presence of persistent hydrocephalus.

The results show that the ratio of the volume of the total brain and the ventricle system is a quick and useful measure that facilitates the identification of patients with persistent hydrocephalus. Postoperative deterioration due to hydrocephalus could be prevented by temporary cerebrospinal fluid drainage with an EVD system perioperatively or by selecting at-risk patients for early VP surgery.

A total of 12 patients (12/169; 16.9%) were indicated for permanent CSF diversion following surgery. Of these, four had preoperative HCP. Although there are studies reporting similar frequencies of postoperative HCP [16], the range largely varies between 3.7 and 42% [5,6,7,8,9] depending on the timing of the evaluation for HCP and the potential under-reporting of HCP cases in the literature focusing on the surgical outcomes and complications of VS surgery.

In a systematic review, Russo et al. [16] demonstrated that tumor size and CSF are predictors of persistent hydrocephalus after VS surgery. Smaller tumors are more likely to be found in shunt-dependent patients [4,16]. Gerganov et al. [10] outlined a higher rate of HCP resolution in patients with smaller VS (3.4 cm vs. 4.3 cm), although this finding lacked statistical significance. Our study could not confirm those findings, although volumetric data collection is less prone to incorrect measurements than diameter measurements [17]. However, more large-scale cohorts with volumetric data are needed to show exact correlations.

In this study, ROC analyses of the VTB ratio 0.712 (95% CI: 0.51–0.91, *p* = 0.036) showed significant results in predicting persistent HCP. The VTB ratio had an optimal cut-off of ≥0.449 with a sensitivity of 67%, a specificity of 81%, and a Youden´s Index of 0.48. ROC analysis for the ventricle system did not show significant results, with an AUC of 0.689 (95% CI: 0.486–0.892, *p* = 0.068). A comparison with other studies was not possible, as no other study had used volumetric data, except for tumor volume, as possible predictors of persistent HCP. The implications of a higher specificity versus a lower sensitivity, particularly in a clinical setting, should be discussed in terms of minimizing unnecessary VP procedures. A high specificity ensures that most patients identified as not needing VP surgery are correctly diagnosed, reducing the risk of over-treatment by using the VTB ratio. The limitation of lower sensitivity implies that certain patients who may benefit from early VP shunt placement could remain unidentified, leading to a delay in necessary interventions. Clinicians are required to balance the risks associated with false positives and false negatives. In scenarios where the implications of untreated hydrocephalus are particularly severe, a lower specificity might be considered acceptable to ensure that high-risk patients receive prompt treatment. Therefore, employing a multifactorial approach that integrates the VTB ratio with additional diagnostic markers could potentially refine clinical decision making and improve patient outcomes.

Multivariable logistic regression analyses of imaging data (pre- and postoperative VS volume, VTB ratio, EoR (%), and patient-specific factors (preoperative hydrocephalus)) revealed that an increased VTB ratio (≥0.049, OR: 6.2, 95% CI: 1.0–38.0, *p* = 0.047) and a low EoR % (<96.4%, OR: 9.1, 95% CI: 1.2–69.3, *p* = 0.032) were independently associated with postoperative VP shunt dependency.

Removing the primary tumor remains the best treatment to reduce the risk of postoperative persistent hydrocephalus, achieving a resolution in more than 85% of cases [16]. However, a high preoperative VTB ratio is an independent predictor for the requirement for postoperative VP shunt surgery, and patients may therefore benefit from perioperative EVD placement. Since VS-associated hydrocephalus is theoretically reversible, the latter option should be considered before VP surgery [10].

This study presents several considerations that warrant acknowledging. A potential limitation of this study is its retrospective design, which means that it is not possible to completely rule out bias, incomplete data, or inconsistent and limited follow-up time. Additionally, a larger sample size would provide more robust statistical analyses.

It should be noted that this study applied highly selective inclusion criteria, such as homogeneous treatment via the retrosigmoid approach in the semi-sitting positioning, and sporadic VSs only. Vychopen et al. showed, in a meta-analysis with 736 patients, that the semi-sitting position has no effect on the development of hydrocephalus in VS compared to the park-bench position [22]. Other studies have demonstrated that volumetric measurement is superior to two-dimensional measurement of VS in terms of accuracy and reliability of outcome assessments [20]. Furthermore, the evaluation of VP shunt dependency is limited by the absence of CSF protein data because perioperative EVD insertion for VS surgery is not performed [16] in the routine at the present institution.

To the best of our knowledge, the present study is the first to investigate hydrocephalus after VS surgery with volumetric MRI data.

## 5. Conclusions

The VTB ratio is an independent predictor for identifying VP-shunt-dependent patients after VS surgery and might be a future quick-to-use clinical tool for preoperative planning for VS surgery. The VTB ratio may serve as a preoperative indicator for neurosurgeons during preoperative assessment to determine the necessity of perioperative EVD placement.

## Figures and Tables

**Figure 1 jcm-13-05789-f001:**
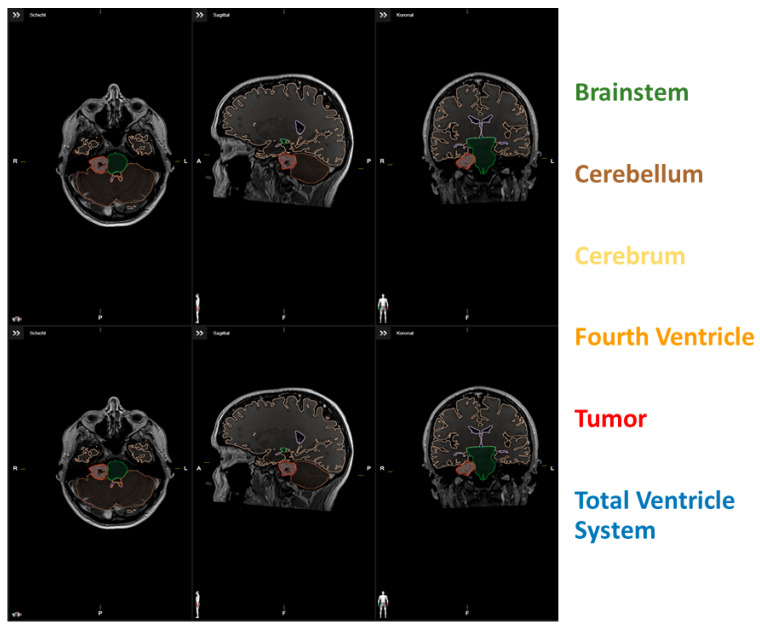
(**upper row**) Segmentation data of a patient with low VTB-ratio; (**lower row**) Segmentation data of a patient with high VTB-ratio.

**Figure 2 jcm-13-05789-f002:**
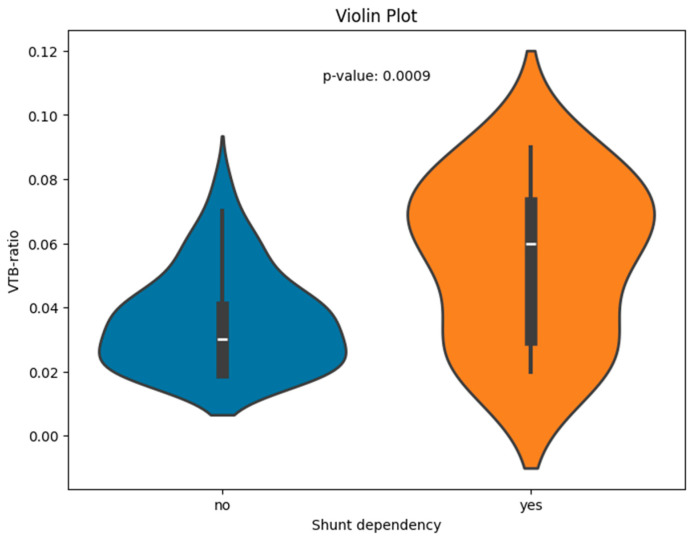
Violin plot of the VTB ratio for patients with and without shunt dependency.

**Figure 3 jcm-13-05789-f003:**
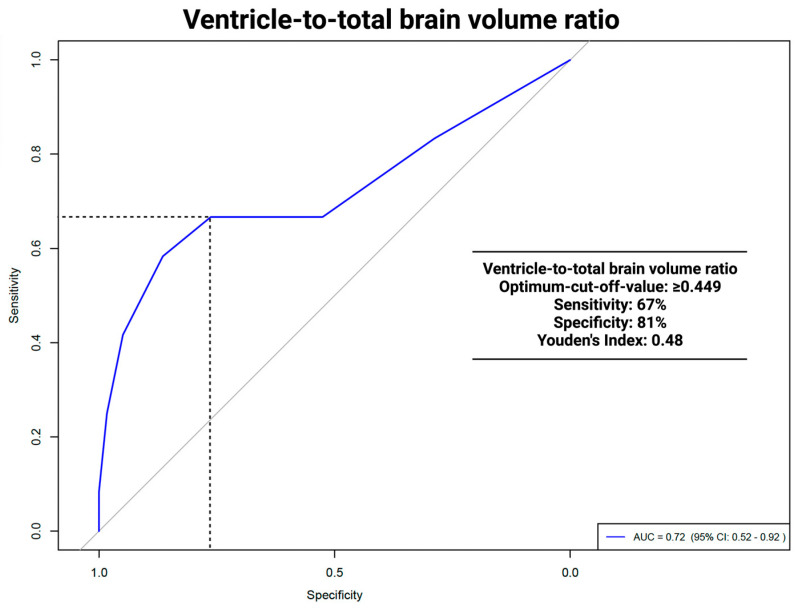
ROC analysis: ventricle to total brain ratio and VP shunt dependencies.

**Figure 4 jcm-13-05789-f004:**
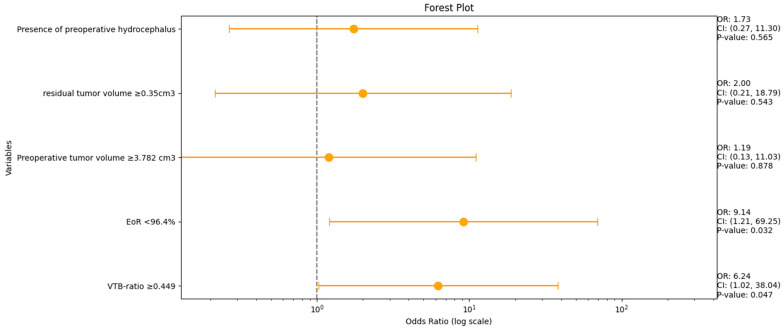
Forest plots from multivariable binary logistic regression analysis: VTB ratio and EoR (%) are independent predictors of persistent HCP in VS.

**Table 1 jcm-13-05789-t001:** Baseline patients’ characteristics.

Characteristics	*n* = 71
Age (mean +/− SD)	59 years (+/−15.5 years)
Sex	56.3% women vs. 43.7% men
VS volume (mean +/− SD)	3.54 +/− 12.39 cm^3^
Preoperative hydrocephalus	4 (5.6%)
Peritumoral brain edema (mean +/− SD)	0 +/− 1.27 cm^3^
Koos grade 1	1 (1.4%)
Koos grade 2	13 (18.3%)
Koos grade 3	17 (23.9%)
Koos grade 4	40 (56.3%)
MIB-I Index (%)	3.0 +/− 1.24
Preoperative V cranial nerve dysfunction	14 (19.7%)
Preoperative VII cranial nerve dysfunction	14 (19.7%)
Preoperative VIII cranial nerve dysfunction	71 (100%)
Preoperative hypacusis	59 (83.1%)
Preoperative anacusis	12 (16.9%)
Preoperative IX cranial nerve dysfunction	4 (5.6%)

Abbreviations: VS: vestibular schwannoma; MIB-I: Molecular Immunology Borstel I.

**Table 2 jcm-13-05789-t002:** Patient characteristics and univariate analysis for shunt-dependency and no-shunt-dependency groups using Fisher’s exact test (two-sided), Spearman correlation, and independent *t*-test.

Characteristics	Shunt (*n* = 12)	No Shunt (*n* = 59)	*p*-Value
Median age (years +/− SD)	57.5 +/− 11.4	56.8 +/− 17.3	0.445
Male sex	3 (25%)	28 (47.5%)	0.208
Median volume of VS, cm^3^			
<3.78	4 (36.4%)	32 (54.2%)	0.336
≥3.78	7 (63.6%)	27 (45.8%)	0.336
Ventricle system, cm^3^			
<52.15	4 (33.3%)	48 (81.4%)	0.002
≥52.15	8 (66.7%)	11 (18.6%)	
Cerebellum, cm^3^			
≤110.5	9 (75%)	12 (20.3%)	<0.001
>110	3 (25%)	47 (79.7%)	
Peritumoral edema (+/−SD), cm^3^	0.0 +/− 0.0	0.31 +/− 1.42	0.251
Brainstem (+/−SD), cm^3^	28.81 +/− 2.9	29.76 +/− 4.03	0.366
Fourth ventricle (+/−SD), cm^3^	1.11 +/− 1.10	0.9+/− 0.54	0.366
Cerebrum (+/−SD), cm^3^	879.5 +/− 51.67	893.21 +/− 96.16	0.066
EoR (%) (+/−SD)	81.49 +/− 28.344	81.22 +/− 17.60	0.146
VTB ratio (+/−SD), cm^3^	0.05 +/− 0.025	0.04 +/− 0.015	<0.001
Evans index (+/−SD)	0.29 +/− 0.54	0.27 +/− 0.03	0.129
Surface area (mean ± SD), mm^2^	2398.69 +/− 1949.18	2180.011 +/− 2242.89	0.946
BMI	24.44 +/− 3.71	26.56 +/− 4.24	0.821
Serum CRP preoperative (mg/L)	2.20 +/− 1.3	2.75 +/− 3.5	0.145
Koos grade			
1	0 (0%)	1 (1.7%)	0.191
2	0 (0%)	13 (22%)	
3	5 (41.7%)	12 (20.3%)	
4	7 (58.3%)	33 (55.9%)	

Abbreviations: VS: vestibular schwannoma; EoR: extent of resection; VTB ratio: ventricle–total brain ratio; BMI: body mass index; CRP: c-reactive protein.

## Data Availability

Research data will not be shared.

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
