# Peer review of "The Ratio of Baseline Ventricle Volume to Total Brain Volume Predicts Postoperative Ventriculo-Peritoneal Shunt Dependency after Sporadic Vestibular Schwannoma Surgery"

_jcm, 2024, doi:10.3390/jcm13195789_

Round 1
Reviewer 1 Report
Comments and Suggestions for Authors
At the beginning of the review, I wanted to congratulate the team of authors on a very interesting article. In fact, at the time of review, this is the first paper that investigates hydrocephalus after VS surgery with volumetric MRI data.
Introduction, material and methods and results presented clearly. I have a few minor comments in this regard I kindly ask the authors to clarify them:
1. In Table 2, median age was inserted twice?
2. Why male / female population presentation is different? Table vs. plot.
3. No expansion of some abbreviations used for the first time
Minor technical issues: Figure 1 seems a bit blurred can You please provide a better resolution?
The discussion carried out correctly, the authors clearly indicate the limitations and the need for larger studies on larger population groups.
Literature seems a bit outdated ~63% is older than 10 years is it possible to look for more recent publications?
In the authors' experience and practice, have there been pregnant patients with VS? If so, what was the management?
Overall, the article is very interesting, presenting the possibility of introducing new tools for treatment planning, but also for patients' counseling and providing information - something I personally missed in the conclusion. Once again, congratulations on the article.
Comments on the Quality of English LanguageMinor linguistic corrections, punctuation to be improved in the final version of the manuscript.
Author Response
Response to Reviewer 1 Comments
- Summary
Thank you very much for reading our manuscript and critically reviewing it. It will help us improve it to a better scientific level and make it more understandable for the readership.
We agree with the reviewer regarding the lack of existent research using volumetric MRI data for investigation of hydrocephalus after VS surgery.
Please find the detailed responses below, and the corresponding revisions in the resubmitted files.
- Point-by-point response to Comments and Suggestions for Authors
Comment 1: In Table 2, median age was inserted twice?
Response 1: Thank you for pointing this out. We agree with this comment and corrected the duplication of the median age in Table 2. The revised version of Table 2 can be found on page [6].
Comment 2: Why male / female population presentation is different? Table vs. plot.
Response 2: We appreciate your observation. We have now standardized the presentation of male/female population data across both the table and corresponding text. This change has been applied to Figure [2] and paragraph [3.2.], both located on page [5], paragraph [3.2.].
Comment 3: No expansion of some abbreviations used for the first time. Figure 1 seems a bit blurred. Can you please provide a better resolution?
Response 3: We have now expanded all abbreviations used for the first time to ensure clarity. These changes can be found throughout the manuscript, and the first occurrences have been updated accordingly on page [2], paragraph [1]. We have also replaced Figure 1 with a higher resolution version. The updated figure is now located on page [2].
Comment 4: The literature seems outdated (~63% older than 10 years). Is it possible to look for more recent publications?
Response 4: Thank you for this suggestion, indeed we have updated the literature by including more recent references from the past 10 years. The updated references can be found first on page [2], paragraph [1].
Comment 5: In the authors' experience and practice, have there been pregnant patients with VS? If so, what was the management?
Response 5: That would be an interesting point. Unfortunately, we are unable to make a statement on pregnant VS patients because in our institutional cohort no pregnant patients underwent VS surgery.
- Response to Comments on the Quality of English Language
Point 1: Minor linguistic corrections, punctuation to be improved in the final version of the manuscript.
Response 1: We have revised the text to enhance clarity and ensure grammatical accuracy. All changes related to the language can be found throughout the manuscript in track changes.
- Additional Clarifications
None at this time.
Reviewer 2 Report
Comments and Suggestions for Authors
For the author
The study explores factors influencing the necessity of ventriculoperitoneal (VP) shunt surgery following surgery for sporadic vestibular schwannoma (VS) in cases of persistent obstructive hydrocephalus. MRI volumetric analyses were conducted pre- and postoperatively on 71 patients to assess tumor volumes, brain structures, and peritumoral edema. The study introduces the ventricle-to-total brain volume ratio (VTB-ratio) as a significant predictor of postoperative VP-shunt dependency, with an AUC of 0.71, and a sensitivity and specificity of 67% and 81%, respectively, for the threshold value ≥0.449. Multivariate logistic regression identified an increased VTB-ratio (≥0.049) and a lesser extent of resection (EoR < 96.4%) as independent predictors of VP-shunt dependency. The findings suggest that primary tumor removal is crucial, but patients with a high preoperative VTB-ratio might benefit from perioperative EVD placement to mitigate the risk of postoperative hydrocephalus. Before considering acceptance, it is recommended that the author makes the following revisions.
1. The author utilized the brain ventricles to total brain volume (VTB-ratio) for assessment. However, it should be noted that in cases with preoperative hydrocephalus (four cases in this cohort), the VTB-ratio may increase compare to others. Does this measurement account for the volume occupied by the tumor itself? Additionally, were these patients with preoperative hydrocephalus subjected to preoperative V-P shunting? Clarification on these points is necessary to ensure the accuracy and relevance of the VTB-ratio in the study's context.
2. In Clinical Data Recording (line84-92) and table 2, it is suggested that specific factors such as brainstem displacement and cerebellopontine angle (CPA) diameter be considered. These values are clinically more indicative of the risk of cerebrospinal fluid (CSF) leakage and should be taken into account to balance the risk profile of the cohort before assessing the dependency on V-P shunting. Including these parameters will enhance the robustness of the study's findings and their clinical applicability.
3. In Table 2, the inclusion of gender and peritumoral edema may not be necessary for the analysis. The cutoff values should be clearly explained in terms of their significance for grouping, such as "ventricle < 52.15" and "cerebellum < 110.5". Please provide detailed descriptions of these metrics in the methods section, including the rationale behind the chosen cutoffs and the measurement protocols for "ventricle system," "fourth ventricle," "cerebrum," and "surface area." This clarification will aid in understanding the methodology and the basis for the categorization of patients within the study.
4. In Figure 3 (line201), the sensitivity is 67% and the specificity as 81%. This suggests that the assessment method is more effective at identifying patients who will require VP shunting ("roll in"), but its sensitivity in determining which patients will not require VP shunting appears to be less pronounced. The discussion should address this point, exploring the implications of these findings for the clinical utility of the assessment tool. Specifically, the discussion should consider how the balance between sensitivity and specificity affects the tool's ability to accurately predict both the need for and the absence of the need for VP shunting in patients.
5. In section 3.3 (line203) and figure 4 (line212), persistent hydrocephalus indicates that only the extent of resection (EoR) has discriminative power. It is suggested that a more comprehensive evaluation is necessary, as relying solely on preoperative factors may be insufficient. It is recommended that the 12 cases of VP shunting be further stratified to analyze surgery-related factors, such as whether the internal auditory canal was opened, whether the mastoid air cells were violated, and whether repair was performed. Additionally, from a clinical perspective, gender and age are not related to tumor characteristics and should not be included in the analysis. The residual tumor volume should be analyzed in conjunction with the preoperative tumor volume and the percentage of residual, as assessing the residual volume in isolation does not provide meaningful insights. Including these considerations will enhance the depth and relevance of the study's findings.
6. In section 3.2 (lines 161-162), the author states that 12 patients required postoperative VP-shunt insertion, yet in the conclusion (lines 30, 220, 274), there are multiple references to the assessment to determine the necessity of a perioperative EVD placement. It is important to clarify whether the author understands the distinction between postoperative VP-shunt insertion and perioperative EVD placement. Please confirm if this is a misstatement and provide the necessary corrections to ensure the accuracy and consistency of the manuscript's findings and conclusions.
7. It has been established in the literature that larger ventricles (Evans ratio ≥0.4) and a higher cystic portion are indicative of persistent hydrocephalus (ref.1). Furthermore, severe hydrocephalus, cystic tumors, and the extent of resection have been linked to treatment failure for hydrocephalus. Given that the hydrocephalus treatment failure rate was highest in the EVD group among the treatments analyzed, including VP shunting, ETV, and EVD, it is essential to determine whether the author has thoroughly evaluated the comparative predictive value of the methods proposed in this paper against Evans' criteria and the therapeutic value of the VP shunt treatment in comparison to ETV and EVD.
8. Although the author notes an inability to assess "CSF protein data" (line 263), it is crucial to understand whether postoperative evaluations of ventricular size and volume changes were conducted. Given that the type of hydrocephalus diagnosed preoperatively may differ postoperatively, such assessments could significantly guide the predictive value of diagnostics and the selection of treatment protocols. Typically, in patients with acoustic neuromas and hydrocephalus, communicating hydrocephalus is more common than obstructive hydrocephalus, and these patients may not benefit from VP shunting, ETV, or EVD as much (ref2,3).
9. In addition, it is of interest that the author's institution employs "semi-sitting positioning," whereas "park bench position" is more commonly reported in the literature. It would be beneficial for the author to assess whether the choice of positioning has any correlation with the occurrence of postoperative hydrocephalus.
10. others revision: page1-line15, "HCP" should be spelled out in full as "hydrocephalus, HCP", the first occurrence of abbreviations should be accompanied by their full names.
references
1. Shin DW, Song SW, Chong S, Kim YH, Cho YH, Hong SH, Kim JH. Treatment Outcome of Hydrocephalus Associated with Vestibular Schwannoma. J Clin Neurol. 2021 Jul;17(3):455-462. doi: 10.3988/jcn.2021.17.3.455. PMID: 34184454; PMCID: PMC8242310.
2. Nakahara M, Imahori T, Sasayama T, Nakai T, Taniguchi M, Komatsu M, Kanzawa M, Kohmura E. Refractory communicating hydrocephalus after radiation for small vestibular schwannoma with asymptomatic ventriculomegaly: A case report. Radiol Case Rep. 2020 May 15;15(7):1023-1028. doi: 10.1016/j.radcr.2020.04.063. PMID: 32435322; PMCID: PMC7229413.
3. Cauley KA, Ratkovits B, Braff SP, Linnell G. Communicating hydrocephalus after gamma knife radiosurgery for vestibular schwannoma: an MR imaging study. AJNR Am J Neuroradiol. 2009 May;30(5):992-4. doi: 10.3174/ajnr.A1379. Epub 2008 Nov 27. PMID: 19039040; PMCID: PMC7051663.
Comments on the Quality of English Language
the English expression of the article needs improvement in some parts to meet the standards of academic writing.
Author Response
Response to Reviewer 2 Comments
- Summary
Thank you for your detailed and insightful feedback. We have carefully reviewed all points and made revisions accordingly. Please find our responses below, all changes are highlighted in the revised manuscript.
- Point-by-point response to Comments and Suggestions for Authors
Comment 1:
The author utilized the brain ventricles to total brain volume (VTB-ratio) for assessment. However, it should be noted that in cases with preoperative hydrocephalus (four cases in this cohort), the VTB-ratio may increase compare to others. Does this measurement account for the volume occupied by the tumor itself? Additionally, were these patients with preoperative hydrocephalus subjected to preoperative V-P shunting? Clarification on these points is necessary to ensure the accuracy and relevance of the VTB-ratio in the study's context.
Response 1: Thank you for raising this important point. We acknowledge that preoperative hydrocephalus could influence the VTB-ratio. In our analysis, we accounted for the tumor volume by not including it in the measurement of the total brain volume in cases. Additionally, none of the patients with preoperative hydrocephalus underwent preoperative VP shunting. This clarification has been added to the manuscript on page [3], paragraph [2.3.] and page [4], paragraph [3.1.]
Comment 2: In Clinical Data Recording (line 84-92) and table 2, it is suggested that specific factors such as brainstem displacement and cerebellopontine angle (CPA) diameter be considered. These values are clinically more indicative of the risk of cerebrospinal fluid (CSF) leakage and should be taken into account to balance the risk profile of the cohort before assessing the dependency on V-P shunting. Including these parameters will enhance the robustness of the study's findings and their clinical applicability.
Response 2: We agree with the suggestion to include brainstem displacement and CPA diameter. We added the Koos grade in table 2 in our analysis on page [6], table [2]. Tumor size (in CPA) is addressed by volumetric measurement.
Comment 3: In Table 2, the inclusion of gender and peritumoral edema may not be necessary for the analysis. The cutoff values should be clearly explained in terms of their significance for grouping, such as "ventricle < 52.15" and "cerebellum < 110.5". Please provide detailed descriptions of these metrics in the methods section, including the rationale behind the chosen cutoffs and the measurement protocols for "ventricle system," "fourth ventricle," "cerebrum," and "surface area." This clarification will aid in understanding the methodology and the basis for the categorization of patients within the study.
Response 3: We appreciate your feedback. We provided detailed descriptions of the cutoff values on page [4], paragraph [2.6.]. We have also clarified the measurement protocols for the ventricle system, fourth ventricle, cerebrum, and surface area. These updates can be found in the methods section on page [3], paragraph [2.3.].
Peritumoral edema has only shown non-significant results and is therefore only listed in table 2 and not further discussed. We think it is essential to include gender in the analysis of brain tumor treatments, as there is strong evidence showing significant differences in outcomes, treatment responses, and molecular pathways between men and women [1-4].
Comment 4: In Figure 3 (line201), the sensitivity is 67% and the specificity as 81%. This suggests that the assessment method is more effective at identifying patients who will require VP shunting ("roll in"), but its sensitivity in determining which patients will not require VP shunting appears to be less pronounced. The discussion should address this point, exploring the implications of these findings for the clinical utility of the assessment tool. Specifically, the discussion should consider how the balance between sensitivity and specificity affects the tool's ability to accurately predict both the need for and the absence of the need for VP shunting in patients.
Response 4: Thank you for this suggestion. We have expanded the discussion to include a more in-depth analysis of the sensitivity and specificity of the VTB-ratio as a predictive tool for VP shunting. The implications of a higher specificity versus lower sensitivity, particularly in a clinical setting, have been discussed on page [9], paragraph [4].
Comment 5: In section 3.3 (line203) and figure 4 (line212), persistent hydrocephalus indicates that only the extent of resection (EoR) has discriminative power. It is suggested that a more comprehensive evaluation is necessary, as relying solely on preoperative factors may be insufficient. It is recommended that the 12 cases of VP shunting be further stratified to analyze surgery-related factors, such as whether the internal auditory canal was opened, whether the mastoid air cells were violated, and whether repair was performed. Additionally, from a clinical perspective, gender and age are not related to tumor characteristics and should not be included in the analysis. The residual tumor volume should be analyzed in conjunction with the preoperative tumor volume and the percentage of residual, as assessing the residual volume in isolation does not provide meaningful insights. Including these considerations will enhance the depth and relevance of the study's findings.
Response 5: Thank you for the valuable feedback. Our multivariable analysis showed that both VTB-ratio and extent of resection are independent factors being associated with postoperative hydrocephalus. We will address the following points in the revised manuscript. Gender and Age are removed from analysis on page [8], paragraph [3.3.] and Figure [4]. In our current analysis, we have indeed taken the preoperative tumor volume into account by incorporating a relative measure, specifically the extent of resection (EoR). This approach allows us to evaluate the residual tumor volume in relation to the initial tumor size, providing a more meaningful and comprehensive analysis. By focusing on the preoperative volume residual tumor volume and the EoR (%), we address the concern of assessing residual tumor volume in isolation, ensuring more relevant insights into postoperative outcomes.
Comment 6: In section 3.2 (lines 161-162), the author states that 12 patients required postoperative VP-shunt insertion, yet in the conclusion (lines 30, 220, 274), there are multiple references to the assessment to determine the necessity of a perioperative EVD placement. It is important to clarify whether the author understands the distinction between postoperative VP-shunt insertion and perioperative EVD placement. Please confirm if this is a misstatement and provide the necessary corrections to ensure the accuracy and consistency of the manuscript's findings and conclusions.
Response 6: Thank you for pointing this out. We have clarified the distinction between postoperative VP shunt insertion and perioperative EVD placement. In our cohort, no patient received a perioperative EVD device. However, this is a known procedure from the literature [6,7] . Therefore, we found it appropriate to address this topic in the discussion, as the usefulness and applicability of our volumetrics based ratio could determine in which patients a perioperative EVD might be useful, as there is a risk of hydrocephalus. We have reorganized the discussion on page [8], paragraph [4] to make this approach much clearer.
Comment 7: It has been established in the literature that larger ventricles (Evans ratio ≥0.4) and a higher cystic portion are indicative of persistent hydrocephalus (ref.1). Furthermore, severe hydrocephalus, cystic tumors, and the extent of resection have been linked to treatment failure for hydrocephalus. Given that the hydrocephalus treatment failure rate was highest in the EVD group among the treatments analyzed, including VP shunting, ETV, and EVD, it is essential to determine whether the author has thoroughly evaluated the comparative predictive value of the methods proposed in this paper against Evans' criteria and the therapeutic value of the VP shunt treatment in comparison to ETV and EVD.
Response 7: We appreciate this recommendation. The Evans index is commonly used to measure the width of the lateral ventricles and support a hydrocephalus diagnosis. This method is particularly common in patients with normal pressure hydrocephalus (NPH).
However, vestibular schwannomas grow into the cerebellopontine angle (CPA), where the Evans index is usually irrelevant as it focuses on the ventricle size in the brain and not on lesions in the cerebellopontine angle.
Comment 8: Although the author notes an inability to assess "CSF protein data" (line 263), it is crucial to understand whether postoperative evaluations of ventricular size and volume changes were conducted. Given that the type of hydrocephalus diagnosed preoperatively may differ postoperatively, such assessments could significantly guide the predictive value of diagnostics and the selection of treatment protocols. Typically, in patients with acoustic neuromas and hydrocephalus, communicating hydrocephalus is more common than obstructive hydrocephalus, and these patients may not benefit from VP shunting, ETV, or EVD as much (ref2,3).
Response 8:
Thank you very much for your thoughtful and constructive feedback on our manuscript. We greatly appreciate your insights, particularly regarding the postoperative evaluation of ventricular size and volume changes.
The measurement of a scientific reliable perioperative delta of the volumetrics is not be possible in our institution because we perform no routine early postoperative MRIs (<48 h such as in glioma surgery) in VS patients. VS patients undergo their first postoperative MR-imaging at 3-months after surgery. Hence, a scientific reliable MR-volumetric assessment of the delta/mean differences of MR-volumetrics would necessitate a routine postoperative MR-imaging of all VS patients. In case of clinical deterioration the patients undergo postoperative CT scans. However, in the planning of the study we also considered to determine the delta but decided that it would not be appropriate to use preoperative MRI data and postoperative CT data. The use of volumetrics from different scanning procedures would not be scientifically reliable.
Comment 9: In addition, it is of interest that the author's institution employs "semi-sitting positioning," whereas "park bench position" is more commonly reported in the literature. It would be beneficial for the author to assess whether the choice of positioning has any correlation with the occurrence of postoperative hydrocephalus.
Response 9: We appreciate your feedback. In a recent meta-analysis by Vychopen et al. [5] it has been demonstrated that the semi-sitting position has no effect on the development of hydrocephalus. In particular, see Vychopen et al., [Section 3.5.2 and Figure 7], who has pooled data from 736 patients and no difference regarding presence of hydrocephalus among patients in the semi-sitting or park-bench position was observed. This has been added in the discussion on page [10], paragraph [4] to clarify it for the reader.
Comment 10: others revision: page1-line15, "HCP" should be spelled out in full as "hydrocephalus, HCP", the first occurrence of abbreviations should be accompanied by their full names.
Response 10: We have ensured that all abbreviations, including "HCP," are spelled out in full upon their first use. This correction can be found on page [1].
- Response to Comments on the Quality of English Language
Minor grammatical and structural improvements have been made throughout the manuscript to enhance readability.
[1] Jovanovich N, Habib A, Chilukuri A, Hameed NUF, Deng H, Shanahan R, Head JR, Zinn PO. Sex-specific molecular differences in glioblastoma: assessing the clinical significance of genetic variants. Front Oncol. 2024 Jan 23;13:1340386. doi: 10.3389/fonc.2023.1340386. PMID: 38322284; PMCID: PMC10844554.
[2] Sun T, Plutynski A, Ward S, Rubin JB. An integrative view on sex differences in brain tumors. Cell Mol Life Sci. 2015 Sep;72(17):3323-42. doi: 10.1007/s00018-015-1930-2. Epub 2015 May 19. PMID: 25985759; PMCID: PMC4531141.
[3] Lee J, Kay K, Troike K, Ahluwalia MS, Lathia JD. Sex Differences in Glioblastoma Immunotherapy Response. Neuromolecular Med. 2022 Mar;24(1):50-55. doi: 10.1007/s12017-021-08659-x. Epub 2021 Apr 17. PMID: 33864598.
[4] Jang B, Yoon D, Lee JY, Kim J, Hong J, Koo H, Sa JK. Integrative multi-omics characterization reveals sex differences in glioblastoma. Biol Sex Differ. 2024 Mar 16;15(1):23. doi: 10.1186/s13293-024-00601-7. PMID: 38491408; PMCID: PMC10943869.
[5] Vychopen M, Arlt F, Güresir E, Wach J. How to position the patient? A meta-analysis of positioning in vestibular schwannoma surgery via the retrosigmoid approach. Front Oncol. 2023 Feb 1;13:1106819. doi: 10.3389/fonc.2023.1106819. PMID: 36816965; PMCID: PMC9929142.
[6] Huang X, Xu J, Xu M, Chen M, Ji K, Ren J, Zhong P. Functional outcome and complications after the microsurgical removal of giant vestibular schwannomas via the retrosigmoid approach: a retrospective review of 16-year experience in a single hospital. BMC Neurol. 2017 Jan 31;17(1):18. doi: 10.1186/s12883-017-0805-6. PMID: 28137246; PMCID: PMC5282727.
[7] di Russo P, Fava A, Vandenbulcke A, Miyakoshi A, Kohno M, Evins AI, Esposito V, Morace R. Characteristics and management of hydrocephalus associated with vestibular schwannomas: a systematic review. Neurosurg Rev. 2021 Apr;44(2):687-698. doi: 10.1007/s10143-020-01287-2. Epub 2020 Apr 7. PMID: 32266553.
Round 2
Reviewer 2 Report
Comments and Suggestions for Authors
Dear Author,
Thank you for your itemized responses. Most of my questions have been answered, except for the seventh comment/response, where the author believes that the CPA region is unrelated to the size of the ventricles, which is inaccurate. In fact, vestibular schwannoma often cause obstructive hydrocephalus, leading to dilation of the aqueduct of Sylvius and the lateral ventricles, which is particularly common in tumors larger than 4 cm. In fact, some patients present with symptoms of increased intracranial pressure, and surgery will first consider external ventricular drainage to alleviate the discomfort caused by increased intracranial pressure before considering tumor resection. Please clarify this point. The rest of the questions have been well answered. Thank you for the author's efforts.
Reviewer.
Author Response
Dear Reviewer,
Thank you for your constructive feedback and for bringing attention to this important point regarding the relationship between vestibular schwannomas and ventricular dilation. We revised the manuscript to reflect this important consideration and added on page 6, table 2 the results of the Evans-Score. In the comparison between the shunt-dependent and non-shunt-dependent groups, the following results for the Evan’s ratio and other patient characteristics are observed: The mean Evans score for the shunt group is 0.29 +/- 0.54, and for the no shunt group, it is 0.27 +/- 0.03. The p-value for this comparison is 0.129, indicating no statistically significant difference between the two groups in terms of Evan’s score in the present investigation.
We appreciate your insightful comments and believe that this clarification will improve the quality of the manuscript.